# Generalizing Tree Probability Estimation via Bayesian Networks

**Cheng Zhang**
Computational Biology Program
Fred Hutchinson Cancer Research Center
Seattle, WA 98109
chengz23@fredhutch.org

**Frederick A. Matsen IV**
Computational Biology Program
Fred Hutchinson Cancer Research Center
Seattle, WA 98109
matsen@fredhutch.org

## Abstract

Probability estimation is one of the fundamental tasks in statistics and machine learning. However, standard methods for probability estimation on discrete objects do not handle object structure in a satisfactory manner. In this paper, we derive a general Bayesian network formulation for probability estimation on leaf-labeled trees that enables flexible approximations which can generalize beyond observations. We show that efficient algorithms for learning Bayesian networks can be easily extended to probability estimation on this challenging structured space. Experiments on both synthetic and real data show that our methods greatly outperform the current practice of using the empirical distribution, as well as a previous effort for probability estimation on trees.

## 1 Introduction

Leaf-labeled trees, where labels are associated with the observed variables, are extensively used in probabilistic graphical models. A typical example is the phylogenetic leaf-labeled tree, which is the fundamental structure for modeling the evolutionary history of a family of genes [Felsenstein, 2003, Friedman et al., 2002]. Inferring a phylogenetic tree based on a set of DNA sequences under a probabilistic model of nucleotide substitutions has been one of the central problems in computational biology, with a wide range of applications from genomic epidemiology [Neher and Bedford, 2015] to conservation genetics [DeSalle and Amato, 2004]. To account for the phylogenetic uncertainty, Bayesian approaches are adopted [Huelsenbeck et al., 2001] and Markov chain Monte Carlo (MCMC) [Yang and Rannala, 1997, Mau et al., 1999, Huelsenbeck and Ronquist, 2001] is commonly used to sample from the posterior of phylogenetic trees. Posterior probabilities of phylogenetic trees are then typically estimated with simple sample relative frequencies (SRF), based on those MCMC samples.

While classical, this empirical approach is unsatisfactory for tree posterior estimation due to the combinatorially exploding size of tree space. Specifically, SRF does not support trees beyond observed samples (i.e., simply sets the probabilities of unsampled trees to zero), and is prone to unstable estimates for low-probability trees. As a result, reliable estimations using SRF usually require impractically large sample sizes. Previous work [Höhna and Drummond, 2012, Larget, 2013] attempted to remedy these problems by harnessing the similarity among trees and proposed several probability estimators using MCMC samples based on conditional independence of separated subtrees. Although these estimators do extend to unsampled trees, the conditional independence assumption therein is often too strong to provide accurate approximations for posteriors inferred from real data [Whidden and Matsen, 2015].

In this paper, we present a general framework for tree probability estimation given a collection of trees (e.g., MCMC samples) by introducing a novel structure called *subsplit Bayesian networks* (SBNs). This structure provides rich distributions over the entire tree space and hence differs from existing applications of Bayesian networks in phylogenetic inference [e.g. Strimmer and Moulton,

2000, Höhna et al., 2014] to compute tree likelihood. Moreover, SBNs relax the conditional clade independence assumption and allow easy adjustment for a variety of flexible dependence structures between subtrees. They also allow many efficient learning algorithms for Bayesian networks to be easily extended to tree probability estimation. Inspired by weight sharing used in convolutional neural networks [LeCun et al., 1998], we propose *conditional probability sharing* for learning SBNs, which greatly reduces the number of free parameters by exploiting the similarity of local structures among trees. Although initially proposed for rooted trees, we show that SBNs can be naturally generalized to unrooted trees, which leads to a missing data problem that can be efficiently solved through expectation maximization. Finally, we demonstrate that SBN estimators greatly outperform other tree probability estimators on both synthetic data and a benchmark of challenging phylogenetic posterior estimation problems. The SBN framework also works for general leaf-labeled trees, however for ease of presentation, we restrict to leaf-labeled bifurcating trees in this paper.

## 2  Background

A leaf-labeled bifurcating tree is a binary tree (rooted or unrooted) with labeled leaves (e.g., leaf nodes associated with observed variables or a set of labels); we refer to it as a tree for short. Recently, several probability estimators on tree spaces have been proposed that exploit the similarity of *clades*, a local structure of trees, to generalize beyond observed trees. Let $\mathcal{X} = \{O_1, \ldots, O_N\}$ be a set of $N$ labeled leaves. A *clade* $X$ of $\mathcal{X}$ is a nonempty subset of $\mathcal{X}$. Given a rooted tree $T$ on $\mathcal{X}$, one can find its unique clade decomposition as follows. Start from the root, which has a trivial clade that contains all the leaves $C_1$. This clade first splits into two subclades $C_2$, $C_3$. The splitting process continues recursively onto each successive subclade until there are no subclades to split. Finally, we obtain a

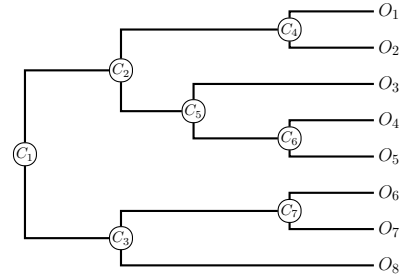

Figure 1: A rooted tree and its clade decomposition. Each clade corresponds to the set of offspring leaves (e.g., $C_2 = \{O_1, O_2, O_3, O_4, O_5\}$ and $C_3 = \{O_6, O_7, O_8\}$).

collection of nontrivial clades $T_{\mathcal{C}}$. As for the tree in Figure 1, $T_{\mathcal{C}} = \{C_2, C_3, C_4, C_5, C_6, C_7\}$. This way, $T$ is represented uniquely as a set of clades $T_{\mathcal{C}}$. Therefore, distributions over the tree space can be specified as distributions over the space of sets of clades. Again, for Figure 1:

$$p(T) = p(T_{\mathcal{C}}) = p(C_2, C_3, C_4, C_5, C_6, C_7) \tag{1}$$

The clade decomposition representation enables distributions that reflect the similarity of trees through the local clade structure. However, a full parameterization of this approach over all rooted trees on $\mathcal{X}$ using rules of conditional probability is intractable even for a moderate $N$. Larget [2013], building on work of Höhna and Drummond [2012], introduced the *Conditional Clade Distribution* (CCD) which assumes that given the existence of an edge in a tree, clades that further refine opposite sides of the edge are independent (see the Supplementary Material (SM) for a more detailed discussion). CCD greatly reduces the number of parameters. For example, (1) has the following CCD approximation

$$p_{\mathrm{ccd}}(T) = p(C_2, C_3)p(C_4, C_5|C_2)p(C_6|C_5)p(C_7|C_3)$$

However, CCD also introduces strong bias which makes it insufficient to capture the complexity of inferred posterior distributions on real data (see Figure 5). In particular, certain clades may depend on their sisters. This motivates a more flexible set of approximate distributions.

## 3  A Subsplit Bayesian Network Formulation

In addition to the clade decomposition representation, a rooted tree $T$ can also be uniquely represented as a set of subsplits. Let $\succ$ be a total order on clades (e.g., lexicographical order). A *subsplit* $(Y, Z)$ of a clade $X$ is an ordered pair of disjoint subclades of $X$ such that $Y \cup Z = X$ and $Y \succ Z$. For example, the tree in Figure 1 corresponds to the following set of nontrivial subsplits

$$T_{\mathcal{S}} = \{(C_2, C_3), (C_4, C_5), (\{O_3\}, C_6), (C_7, \{O_8\})\}$$

with lexicographical order on clades. Moreover, this set-of-subsplits representation of trees inspires a natural probabilistic Bayesian network formulation as follows (Figure 2):

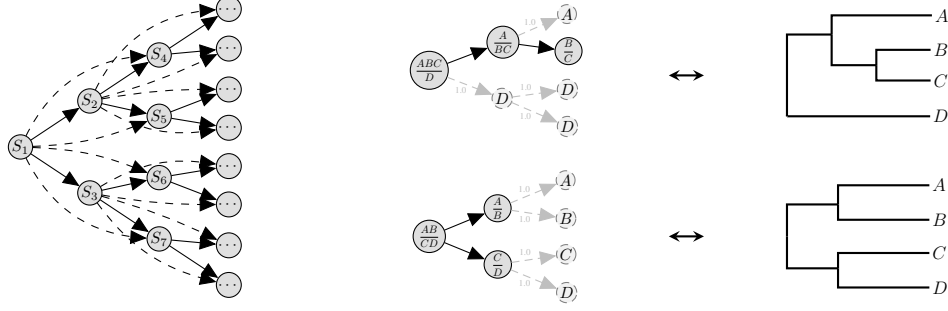

Figure 2: The subsplit Bayesian network formulation. **Left**: A general Bayes net for rooted trees. Each node represents a subsplit-valued or singleton-clade-valued random variable. The solid full and complete binary tree network is $\mathcal{B}^*_{\mathcal{X}}$. **Middle/Right**: Examples of rooted trees with 4 leaves. Note that the *solid dark nets* that represent the true splitting processes of the trees may have dynamical structures. By allowing singleton clades continue to split (the *dashed gray nets*) until depth of 3, both nets grow into the full and complete binary tree of depth 3.

**Definition 1.** *A subsplit Bayesian network (SBN) $\mathcal{B}_{\mathcal{X}}$ on a leaf set $\mathcal{X}$ of size $N$ is a Bayesian network whose nodes take on subsplit values or singleton clade values of $\mathcal{X}$ and: i) has depth $N-1$ (the root counts as depth 1); ii) The root node takes on subsplits of the entire leaf set $\mathcal{X}$; iii) contains a full and complete binary tree network $\mathcal{B}^*_{\mathcal{X}}$ as a subnetwork.*

Note that $\mathcal{B}^*_{\mathcal{X}}$ itself is an SBN and is contained in all SBNs; therefore, $\mathcal{B}^*_{\mathcal{X}}$ is the minimum SBN on $\mathcal{X}$. Moreover, $\mathcal{B}^*_{\mathcal{X}}$ induces a natural indexing procedure for the nodes of all SBNs on $\mathcal{X}$: starting from the root node, which is denoted as $S_1$, for any $i$, we denote the two children of $S_i$ as $S_{2i}$ and $S_{2i+1}$, respectively, until a leaf node is reached. We call the parent nodes in $\mathcal{B}^*_{\mathcal{X}}$ the *natural parents*.

**Definition 2.** *We say a subsplit $(Y, Z)$ is* compatible *with a clade $X$ if $Y \cup Z = X$. Moreover, a singleton clade $\{W\}$ is said to be compatible with itself. With natural indexing, we say a full SBN assignment $\{S_i = s_i\}_{i \geq 1}$ is* compatible *if for any interior node assignment $s_i = (Y_i, Z_i)$ (or $\{W_i\}$), $s_{2i}, s_{2i+1}$ are compatible with $Y_i, Z_i$ (or $\{W_i\}$), respectively. Consider a parent-child pair in an SBN, $S_i$ and $S_{\pi_i}$, where $\pi_i$ denotes the index set of the parent nodes of $S_i$. We say an assignment $S_i = s_i, S_{\pi_i} = s_{\pi_i}$ is* compatible *if it can be extended to a compatible assignment of the SBN.*

**Lemma 1.** *Given an SBN $\mathcal{B}_{\mathcal{X}}$, each rooted tree $T$ on $\mathcal{X}$ can be uniquely represented as a compatible assignment of $\mathcal{B}_{\mathcal{X}}$.*

A proof of Lemma 1 is provided in the SM. Unlike in phylogenies, nodes in SBNs take on subsplit (or singleton clade) values that represent the local topological structure of trees. By including the true splitting processes (e.g., $T_S$) of the trees while allowing singleton clades continue to split ("fake" split) until the whole network reaches depth $N-1$ (see Figure 2), each SBN on $\mathcal{X}$ has a fixed structure which contains the full and complete binary tree as a subnetwork. Note that those fake splits are all deterministically assigned, which means the corresponding conditional probabilities are all one. Therefore, the estimated probabilities of rooted trees only depend on their true splitting processes. With SBNs, we can easily construct a family of flexible approximate distributions on the tree space. For example, using the minimum SBN, (1) can be estimated as

$$p(C_2, C_3)p(C_4, C_5|C_2, C_3)p(C_6|C_4, C_5)p(C_7|C_2, C_3)$$

This approximation implicitly assumes that given the existence of a subsplit, subsplits that further refine opposite sides of this subsplit are independent. Note that CCD can be viewed as a simplification where the conditional probabilities are further approximated as follows

$$p(C_4, C_5|C_2, C_3) \approx p(C_4, C_5|C_2), \quad p(C_6|C_4, C_5) \approx p(C_6|C_5), \quad p(C_7|C_2, C_3) \approx p(C_7|C_3)$$

By including the sister clades in the conditional subsplit probabilities, SBNs relax the conditional clade independence assumption made in CCD and allows for more flexible dependent structures between local components (e.g., subsplits in sister-clades). Moreover, one can add more complicated dependencies between nodes (e.g., dashed arrows in Figure 2(a)) and hence easily adjust SBN formulation to provide a wide range of flexible approximate distributions. For general SBNs, the estimated tree probabilities take the following form:

$$p_{\text{sbn}}(T) = p(S_1) \prod_{i>1} p(S_i|S_{\pi_i}). \tag{2}$$

In addition to the superior flexibility, another benefit of the SBN formulation is that these approximate distributions are all naturally normalized if the conditional probability distributions (CPDs) are *consistent*, as defined next:

**Definition 3.** *We say the conditional probability* $p(S_i|S_{\pi_i})$ *is* consistent *if* $p(S_i = s_i|S_{\pi_i} = s_{\pi_i}) = 0$ *for any incompatible assignment* $S_i = s_i, S_{\pi_i} = s_{\pi_i}$.

**Proposition 1.** *If* $\forall i > 1$, $p(S_i|S_{\pi_i})$ *is consistent, then* $\sum_T p_{\mathrm{sbn}}(T) = 1$.

With Lemma 1, the proof is standard and is given in the SM. Furthermore, the SBN formulation also allows us to easily extend many efficient algorithms for learning Bayesian networks to SBNs for tree probability estimation, as we see next.

## 4  Learning Subsplit Bayesian Networks

### 4.1  Rooted Trees

Suppose we have a sample of rooted trees $\mathcal{D} = \{T_k\}_{k=1}^K$ (e.g., from a phylogenetic MCMC run given DNA sequences). As before, each sampled tree can be represented as a collection of subsplits $T_k = \{S_i = s_{i,k}, \ i \geq 1\}$, $k = 1, \ldots, K$ and therefore has the following SBN likelihood

$$L(T_k) = p(S_1 = s_{1,k}) \prod_{i>1} p(S_i = s_{i,k}|S_{\pi_i} = s_{\pi_i,k}).$$

**Maximum Likelihood**   In this complete data scenario, we can simply use maximum likelihood to learn the parameters of SBNs. Denote the set of all observed subsplits of node $S_i$ as $\mathbb{C}_i$, $i \geq 1$ and the set of all observed subsplits of the parent nodes of $S_i$ as $\mathbb{C}_{\pi_i}$, $i > 1$. Assuming that trees are independently sampled, the complete data log-likelihood is

$$\log L(\mathcal{D}) = \sum_{k=1}^K \left( \log p(S_1 = s_{1,k}) + \sum_{i>1} \log p(S_i = s_{i,k}|S_{\pi_i} = s_{\pi_i,k}) \right)$$

$$= \sum_{s_1 \in \mathbb{C}_1} m_{s_1} \log p(S_1 = s_1) + \sum_{i>1} \sum_{\substack{s_i \in \mathbb{C}_i \\ t_i \in \mathbb{C}_{\pi_i}}} m_{s_i,t_i} \log p(S_i = s_i|S_{\pi_i} = t_i) \tag{3}$$

where $m_{s_1} = \sum_{k=1}^K \mathbb{I}(s_{1,k} = s_1)$, $m_{s_i,t_i} = \sum_{k=1}^K \mathbb{I}(s_{i,k} = s_i, s_{\pi_i,k} = t_i)$, $i > 1$ are the frequency counts of the root splits and parent-child subsplit pairs for each interior node respectively, and $\mathbb{I}(\cdot)$ is the indicator function. The maximum likelihood estimates of CPDs have the following simple closed form expressions in terms of relative frequencies:

$$\hat{p}^{\mathrm{ML}}(S_1 = s_1) = \frac{m_{s_1}}{\sum_{s \in \mathbb{C}_1} m_s} = \frac{m_{s_1}}{K}, \quad \hat{p}^{\mathrm{ML}}(S_i = s_i|S_{\pi_i} = t_i) = \frac{m_{s_i,t_i}}{\sum_{s \in \mathbb{C}_i} m_{s,t_i}}.$$

**Conditional Probability Sharing**   We can use the similarity of local structures to further reduce the number of SBN parameters and achieve better generalization, similar to weight sharing for convolutional nets. Indeed, different trees do share lots of local structures, such as subsplits and clades. As a result, the representations of the trees in an SBN could have the same parent-child subsplit pairs, taken by different nodes (see Figure D.1 in SM). Instead of assigning independent sets of parameters for those pairs at different locations, we can use one set of parameters for each of those shared pairs, regardless of their locations in SBNs. We call this specific setting of parameters in SBNs *conditional probability sharing* (see more on parameter sharing in SM). Compared to standard Bayes nets, this index-free parameterization only needs CPDs for each observed parent-child subsplit pair, dramatically reducing the number of parameters in the model.

Now denote the set of all observed splits of $S_1$ as $\mathbb{C}_r$, and the set of all observed parent-child subsplit pairs as $\mathbb{C}_{\mathrm{ch|pa}}$. The log-likelihood $\log L(\mathcal{D})$ in (3) can be rewritten into

$$\log L(\mathcal{D}) = \sum_{s_1 \in \mathbb{C}_r} m_{s_1} \log p(S_1 = s_1) + \sum_{s|t \in \mathbb{C}_{\mathrm{ch|pa}}} m_{s,t} \log p(s|t)$$

where $m_{s,t} = \sum_{k=1}^K \sum_{i>1} \mathbb{I}(s_{i,k} = s, s_{\pi_i,k} = t)$ is the frequency count of the corresponding subsplit pair $s|t \in \mathbb{C}_{\mathrm{ch|pa}}$. Similarly, we have the maximum likelihood estimates of the CPDs for those parent-child subsplit pairs:

$$\hat{p}^{\mathrm{ML}}(s|t) = \frac{m_{s,t}}{\sum_s m_{s,t}}, \quad s|t \in \mathbb{C}_{\mathrm{ch|pa}}. \tag{4}$$

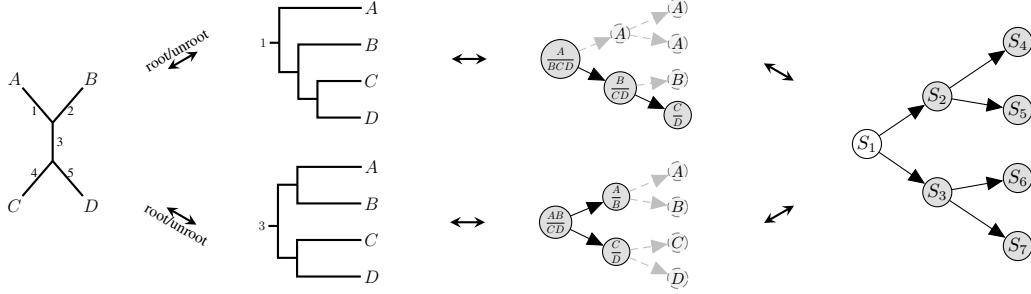

Figure 3: SBNs for unrooted trees. **Left**: A simple four taxon unrooted tree example. It has five edges, $1, 2, 3, 4, 5$, that can be rooted on to make (compatible) rooted trees. **Middle(left)**: Two rooted trees when rooting on edges 1 and 3. **Middle(right)**: The corresponding SBN representations for the two rooted trees. **Right**: An integrated SBN for the unrooted tree with unobserved root node $S_1$.

The main computation is devoted to the collection of the frequency counts $m_{s_1}$ and $m_{s,t}$ which requires iterating over all sampled trees and for each tree, looping over all the edges. Thus the overall computational complexity is $\mathcal{O}(KN)$.

## 4.2 Unrooted Trees

Unrooted trees are commonly used to express undirected relationships between observed variables, and are the most common tree type in phylogenetics. The SBN framework can be easily generalized to unrooted trees because each unrooted tree can be transformed into a rooted tree by placing the root on one of its edges. Since there are multiple possible root placements, each unrooted tree has multiple representations in terms of SBN assignments for the corresponding rooted trees. Unrooted trees, therefore, can be represented using the same SBNs for rooted trees, with root node $S_1$ being unobserved[1] (Figure 3). Marginalizing out the unobserved node $S_1$, we obtain SBN probability estimates for unrooted trees (denoted by $T^{\mathrm{u}}$ in the sequel):

$$p_{\mathrm{sbn}}(T^{\mathrm{u}}) = \sum_{S_1 \sim T^{\mathrm{u}}} p(S_1) \prod_{i>1} p(S_i | S_{\pi_i}) \tag{5}$$

where $\sim$ means all root splits that are compatible with $T^{\mathrm{u}}$. Similar to the SBN approximations for the rooted trees, (5) provides a natural probability distribution over unrooted trees (see a proof in SM).

**Proposition 2.** *Suppose that the conditional probability distributions $p(S_i | S_{\pi_i})$, $i > 1$ are consistent, then* (5) *is a probability distribution over unrooted trees with leaf set $\mathcal{X}$, that is, $\sum_{T^{\mathrm{u}}} p_{\mathrm{sbn}}(T^{\mathrm{u}}) = 1$.*

As before, assume that we have a sample of unrooted trees $\mathcal{D}^{\mathrm{u}} = \{T_k^{\mathrm{u}}\}_{k=1}^K$. Each pair of the unrooted tree and rooting edge corresponds to a rooted tree that can be represented as: $(T_k^{\mathrm{u}}, e) = \{S_i = s_{i,k}^e, i \geq 1\}$, $e \in E(T_k^{\mathrm{u}})$, $1 \leq k \leq K$ where $E(T_k^{\mathrm{u}})$ denotes the edges of $T_k^{\mathrm{u}}$ and $s_{i,k}^e$ denotes all the resulting subsplits when $T_k^{\mathrm{u}}$ is rooted on edge $e$. The SBN likelihood for the unrooted tree $T_k^{\mathrm{u}}$ is

$$L(T_k^{\mathrm{u}}) = \sum_{e \in E(T_k^{\mathrm{u}})} p(S_1 = s_{1,k}^e) \prod_{i>1} p(S_i = s_{i,k}^e | S_{\pi_i} = s_{\pi_i,k}^e).$$

The lost information on the root node $S_1$ means the SBN likelihood for unrooted trees can no longer be factorized. We, therefore, propose the following two algorithms to handle this challenge.

**Maximum Lower Bound Estimates** A simple strategy is to construct tractable lower bounds via variational approximations [Wainwright and Jordan, 2008]

$$LB_q(T^{\mathrm{u}}) := \sum_{S_1 \sim T^{\mathrm{u}}} q(S_1) \left( \log \frac{p(S_1) \prod_{i>1} p(S_i | S_{\pi_i})}{q(S_1)} \right) \leq \log L(T^{\mathrm{u}}) \tag{6}$$

where $q$ is a probability distribution on $S_1 \sim T^{\mathrm{u}}$. In particular, taking $q$ to be uniform on the $2N - 3$ tree edges together with conditional probability sharing gives the *simple average* (SA) lower bound of the data log-likelihood

$$LB^{\mathrm{SA}}(\mathcal{D}^{\mathrm{u}}) := \left( \sum_{s_1 \in \mathbb{C}_r} m_{s_1}^{\mathrm{u}} \log p(S_1 = s_1) + \sum_{s|t \in \mathbb{C}_{\mathrm{ch|pa}}} m_{s,t}^{\mathrm{u}} \log p(s|t) \right) + K \log(2N - 3)$$

**Algorithm 1** Expectation Maximization for SBN

---

**Input:** Data $\mathcal{D}^{\mathrm{u}} = \{T_k^{\mathrm{u}}\}_{k=1}^K$, regularization coeff $\alpha$.
Initialize $\hat{p}^{\mathrm{EM},(0)}$ (e.g., via $\hat{p}^{\mathrm{SA}}$) and $n = 0$. Set equivalent counts $\tilde{m}_{s_1}^{\mathrm{u}}, \tilde{m}_{s,t}^{\mathrm{u}}$ for regularization.
**repeat**

    **E-step.** $\forall\, 1 \le k \le K$, compute $q_k^{(n)}(S_1) = \dfrac{p(T_k^{\mathrm{u}}, S_1 | \hat{p}^{\mathrm{EM},(n)})}{\sum_{S_1 \sim T_k^{\mathrm{u}}} p(T_k^{\mathrm{u}}, S_1 | \hat{p}^{\mathrm{EM},(n)})}$

    **M-step.** Compute the expected frequency counts with conditional probability sharing and update the CPDs by maximizing the regularized $Q$ score

$$\hat{p}^{\mathrm{EM},(n+1)}(S_1 = s_1) = \frac{\overline{m}_{s_1}^{\mathrm{u},(n)} + \alpha \tilde{m}_{s_1}^{\mathrm{u}}}{K + \alpha \sum_{s_1 \in \mathbb{C}_r} \tilde{m}_{s_1}^{\mathrm{u}}}, \quad s_1 \in \mathbb{C}_r, \quad \overline{m}_{s_1}^{\mathrm{u},(n)} = \sum_{k=1}^K \sum_{e \in E(T_k^{\mathrm{u}})} q_k^{(n)}(S_1 = s_1) \mathbb{I}(s_{i,k}^e = s_1)$$

$$\hat{p}^{\mathrm{EM},(n+1)}(s|t) = \frac{\overline{m}_{s,t}^{\mathrm{u},(n)} + \alpha \tilde{m}_{s,t}^{\mathrm{u}}}{\sum_s (\overline{m}_{s,t}^{\mathrm{u},(n)} + \alpha \tilde{m}_{s,t}^{\mathrm{u}})}, \quad s|t \in \mathbb{C}_{\mathrm{ch|pa}}, \quad \overline{m}_{s,t}^{\mathrm{u},(n)} = \sum_{k=1}^K \sum_{e \in E(T_k^{\mathrm{u}})} q_k^{(n)}(S_1 = s_{1,k}^e) \sum_{i>1} \mathbb{I}(s_{i,k}^e = s, s_{\pi_i,k}^e = t)$$

    $n \leftarrow n + 1$
**until** convergence.

---

where

$$m_{s_1}^{\mathrm{u}} = \sum_{k=1}^K \sum_{e \in E(T_k^{\mathrm{u}})} \frac{1}{2N-3} \mathbb{I}(s_{1,k}^e = s_1), \quad m_{s,t}^{\mathrm{u}} = \sum_{k=1}^K \sum_{e \in E(T_k^{\mathrm{u}})} \frac{1}{2N-3} \sum_{i>1} \mathbb{I}(s_{i,k}^e = s, s_{\pi_i,k}^e = t).$$

The maximum SA lower bound estimates are then

$$\hat{p}^{\mathrm{SA}}(S_1 = s_1) = \frac{m_{s_1}^{\mathrm{u}}}{\sum_{s \in \mathbb{C}_r} m_s^{\mathrm{u}}} = \frac{m_{s_1}^{\mathrm{u}}}{K}, \quad s_1 \in \mathbb{C}_r, \quad \hat{p}^{\mathrm{SA}}(s|t) = \frac{m_{s,t}^{\mathrm{u}}}{\sum_s m_{s,t}^{\mathrm{u}}}, \quad s|t \in \mathbb{C}_{\mathrm{ch|pa}}.$$

**Expectation Maximization** The maximum lower bound approximations can be improved upon by adapting the variational distribution $q$, instead of using a fixed one. This, together with conditional probability sharing, leads to an extension of the expectation maximization (EM) algorithm for learning SBNs, which also allows us use of the Bayesian formulation. Specifically, at the E-step of the $n$-th iteration, an adaptive lower bound is constructed through (6) using the conditional probabilities of the missing root node

$$q_k^{(n)}(S_1) = p(S_1 | T_k^{\mathrm{u}}, \hat{p}^{\mathrm{EM},(n)}), \quad k = 1, \dots, K$$

given $\hat{p}^{\mathrm{EM},(n)}$, the current estimate of the CPDs. The lower bound contains a constant term that only depends on the current estimates, and a score function for the CPDs $p$

$$Q^{(n)}(\mathcal{D}^{\mathrm{u}}; p) = \sum_{k=1}^K Q^{(n)}(T_k^{\mathrm{u}}; p) = \sum_{k=1}^K \sum_{S_1 \sim T_k^{\mathrm{u}}} q_k^{(n)}(S_1) \Big( \log p(S_1) + \sum_{i>1} \log p(S_i | S_{\pi_i}) \Big)$$

which is then optimized at the M-step. This variational perspective of the EM algorithm was found and discussed by Neal and Hinton [1998]. The following theorem guarantees that maximizing (or improving) the $Q$ score is sufficient to improve the objective likelihood.

**Theorem 1.** *Let $T^{\mathrm{u}}$ be an unrooted tree. $\forall p$,*

$$Q^{(n)}(T^{\mathrm{u}}; p) - Q^{(n)}(T^{\mathrm{u}}; \hat{p}^{\mathrm{EM},(n)}) \le \log L(T^{\mathrm{u}}; p) - \log L(T^{\mathrm{u}}; \hat{p}^{\mathrm{EM},(n)}).$$

When data is insufficient or the number of parameters is large, the EM approach also easily incorporates regularization [Dempster et al., 1977]. Taking conjugate Dirichlet priors [Buntine, 1991], the regularized score function is

$$Q^{(n)}(\mathcal{D}^{\mathrm{u}}; p) + \sum_{s_1 \in \mathbb{C}_r} \alpha \tilde{m}_{s_1}^{\mathrm{u}} \log p(S_1 = s_1) + \sum_{s|t \in \mathbb{C}_{\mathrm{ch|pa}}} \alpha \tilde{m}_{s,t}^{\mathrm{u}} \log p(s|t)$$

where $\tilde{m}_{s_1}^{\mathrm{u}}, \tilde{m}_{s,t}^{\mathrm{u}}$ are the equivalent sample counts and $\alpha$ is the global regularization coefficient. We then simply maximize the regularized score in the same manner at the M-step. Similarly, this guarantees that the regularized log-likelihood is increasing at each iteration. We summarize the EM

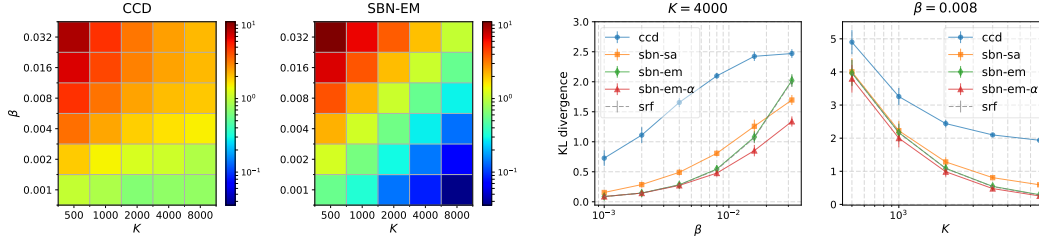

Figure 4: Performance on a challenging tree probability estimation problem with simulated data. **Left:** The KL divergence of CCD and `sbn-em` estimates over a wide range of degree of diffusion $\beta$ and sample size $K$. **Right:** A comparison among different methods for a fixed $K$, as a function of $\beta$ and for a fixed $\beta$, as a function of $K$. Error bar shows one standard deviation over 10 runs.

approach in Algorithm 1. The computational complexities of maximum lower bound estimate and each EM iteration are both $\mathcal{O}(KN)$, the same as CCD and SBNs for rooted trees. See more detailed derivation and proofs in the SM. In practice, EM usually takes several iterations to converge and hence could be more expensive than other methods. However, the gain in approximation makes it a worthwhile trade-off (Table 1). We use the maximum SA lower bound algorithm (`sbn-sa`), the EM algorithm (`sbn-em`) and EM with regularization (`sbn-em-`$\alpha$) in the experiment section.

## 5 Experiments

We compare `sbn-sa`, `sbn-em`, `sbn-em-`$\alpha$ to the classical sample relative frequency (SRF) method and CCD on a synthetic data set and on estimating phylogenetic tree posteriors for a number of real data sets. For all SBN algorithms, we use the simplest SBN, $\mathcal{B}_\mathcal{X}^*$, which we find provide sufficiently accurate approximation in the tree probability estimation tasks investigated in our experiments. For `sbn-em-`$\alpha$, we use the sample frequency counts of the root splits and parent-child subsplit pairs as the equivalent sample counts (see Algorithm 1). The code is made available at `https://github.com/zcrabbit/sbn`.

**Simulated Scenarios** To empirically explore the behavior of SBN algorithms relative to SRF and CCD, we first conduct experiments on a simulated setup. We choose a tractable but challenging tree space, the space of unrooted trees with 8 leaves, which contains 10395 unique trees. The trees are given an arbitrary order. To test the approximation performance on targets of different degrees of diffusion, we generate target distributions by drawing samples from the Dirichlet distributions $\mathrm{Dir}(\beta\mathbf{1})$ of order 10395 with a variety of $\beta$s. The target distribution becomes more diffuse as $\beta$ increases. Simulated data sets are then obtained by sampling from the unrooted tree space according to these target distributions with different sample sizes $K$. The resulting probability estimation is challenging in that the target probabilities of the trees are assigned regardless of the similarity among them. For `sbn-em-`$\alpha$, we adjust the regularization coefficient using $\alpha = \frac{50}{K}$ for different sample sizes. Since the target distributions are known, we use KL divergence from the estimated distributions to the target distributions to measure the approximation accuracy of different methods. We vary $\beta$ and $K$ to control the difficulty of the learning task, and average over 10 independent runs for each configuration. Figure 4 shows the empirical approximation performance of different methods. We see that the learning rate of CCD slows down very quickly as the data size increases, implying that the conditional clade independence assumption could be too strong to provide flexible approximations. On the other hand, `sbn-em` keeps learning efficiently from the data when more samples are available. While all methods tend to perform worse as $\beta$ increases and perform better as $K$ increases, SBN algorithms performs consistently much better than CCD. Compared to `sbn-sa`, `sbn-em` usually greatly improves the approximation with the price of additional computation. When the degree of diffusion is large or the sample size is small, `sbn-em-`$\alpha$ gives much better performance than the others, showing that regularization indeed improves generalization. See the SM for a runtime comparison of different methods with varying $K$ and $\beta$.

**Real Data Phylogenetic Posterior Estimation** We now investigate the performance on large unrooted tree space posterior estimation using 8 real datasets commonly used to benchmark phylogenetic MCMC methods [Lakner et al., 2008, Höhna and Drummond, 2012, Larget, 2013, Whidden and Matsen, 2015] (Table 1). For each of these data sets, 10 single-chain MrBayes [Ronquist et al., 2012] replicates are run for one billion iterations and sampled every 1000 iterations, using the simple Jukes and Cantor [1969] substitution model. We discard the first 25% as burn-in for a total of 7.5 million

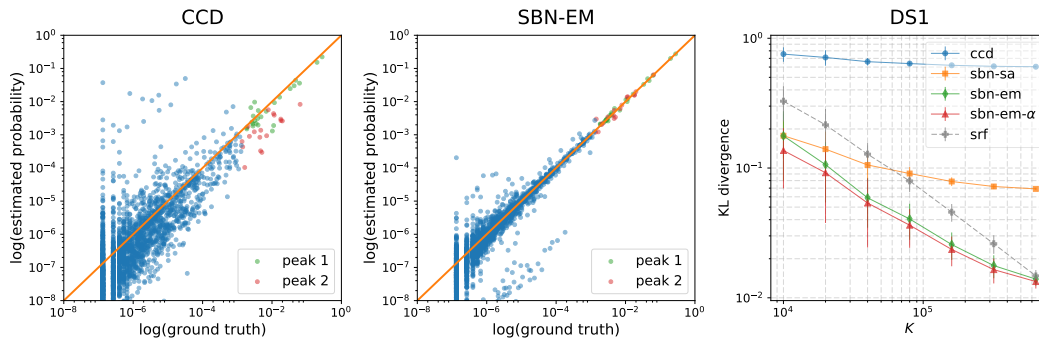

Figure 5: Comparison on DS1, a data set with multiple posterior modes. **Left/Middle:** Ground truth posterior probabilities vs CCD and `sbn-em` estimates. **Right:** Approximation performance as a function of sample size. One standard deviation error bar over 10 replicates.

Table 1: Data sets used for phylogenetic posterior estimation, and approximation accuracy results of different methods across datasets. Sampled trees column shows the numbers of unique trees in the standard run samples. The results are averaged over 10 replicates.

| DATA SET | REFERENCE | (#TAXA, #SITES) | TREE SPACE SIZE | SAMPLED TREES | KL DIVERGENCE TO GROUND TRUTH | | | | |
|---|---|---|---|---|---|---|---|---|---|
| | | | | | SRF | CCD | SBN-SA | SBN-EM | SBN-EM-$\alpha$ |
| DS1 | HEDGES ET AL. [1990] | (27, 1949) | $5.84 \times 10^{32}$ | 1228 | 0.0155 | 0.6027 | 0.0687 | 0.0136 | **0.0130** |
| DS2 | GAREY ET AL. [1996] | (29, 2520) | $1.58 \times 10^{35}$ | 7 | **0.0122** | 0.0218 | 0.0218 | 0.0199 | 0.0128 |
| DS3 | YANG AND YODER [2003] | (36, 1812) | $4.89 \times 10^{47}$ | 43 | 0.3539 | 0.2074 | 0.1152 | 0.1243 | **0.0882** |
| DS4 | HENK ET AL. [2003] | (41, 1137) | $1.01 \times 10^{57}$ | 828 | 0.5322 | 0.1952 | 0.1021 | 0.0763 | **0.0637** |
| DS5 | LAKNER ET AL. [2008] | (50, 378) | $2.84 \times 10^{74}$ | 33752 | 11.5746 | 1.3272 | 0.8952 | 0.8599 | **0.8218** |
| DS6 | ZHANG AND BLACKWELL [2001] | (50, 1133) | $2.84 \times 10^{74}$ | 35407 | 10.0159 | 0.4526 | **0.2613** | 0.3016 | 0.2786 |
| DS7 | YODER AND YANG [2004] | (59, 1824) | $4.36 \times 10^{92}$ | 1125 | 1.2765 | 0.3292 | 0.2341 | 0.0483 | **0.0399** |
| DS8 | ROSSMAN ET AL. [2001] | (64, 1008) | $1.04 \times 10^{103}$ | 3067 | 2.1653 | 0.4149 | 0.2212 | 0.1415 | **0.1236** |

posterior samples per data set. These extremely long "golden runs" form the ground truth to which we will compare various posterior estimates based on standard runs. For these standard runs, we run MrBayes on each data set with 10 replicates of 4 chains and 8 runs until the runs have ASDSF (the standard convergence criteria used in MrBayes) less than 0.01 or a maximum of 100 million iterations. This conservative setting has been shown to find all posterior modes on these data sets [Whidden and Matsen, 2015]. We collect the posterior samples every 100 iterations of these runs and discard the first 25% as burn-in. We apply SBN algorithms, SRF and CCD to the posterior samples in each of the 10 replicates for each data set. For `sbn-em-`$\alpha$, we use $\alpha = 0.0001$ to give some weak regularization[2]. We use KL divergence to the ground truth to measure the performance of all methods.

Previous work [Whidden and Matsen, 2015] has observed that conditional clade independence does not hold in multimodal distributions. Figure 5 shows a comparison on a typical data set, DS1, that has such a "peaky" distribution. We see that CCD underestimates the probability of trees within the subpeak and overestimate the probability of trees between peaks. In contrast, `sbn-em` significantly removes these biases, especially for trees in the 95% credible set.

When applied to a broad range of data sets, we find that SBNs consistently outperform other methods (Table 1). Due to its inability to generalize beyond observed samples, SRF is worse than generalizing probability estimators except for an exceedingly simple posterior with only 7 sampled trees (DS2). CCD is, again, comparatively much worse than SBN algorithms. With weak regularization, `sbn-em-`$\alpha$ gives the best performance in most cases.

To illustrate the efficiency of the algorithms on training data size, we perform an additional study on DS1 with increasing sample sizes and summarize the results in the right panel of Figure 5. As before, we see that CCD slows down quickly while SBN algorithms, especially fully-capable SBN estimators `sbn-em` and `sbn-em-`$\alpha$, keep learning efficiently as the sample size increases. Moreover, SBN algorithms tend to provide much better approximation than SRF when fewer samples are available, which is important in practice where large samples are expensive to obtain.

## 6 Conclusion

We have proposed a general framework for tree probability estimation based on subsplit Bayesian networks. SBNs allow us to exploit the similarity among trees to provide a wide range of flexible probability estimators that generalize beyond observations. Moreover, they also allow many efficient Bayesian network learning algorithms to be extended to tree probability estimation with ease. We report promising numerical results demonstrating the importance of being both flexible and generalizing when estimating probabilities on trees. Although we present SBNs in the context of leaf-labeled bifurcating trees, it can be easily adapted for general leaf-labeled trees by allowing partitions other than subsplits (bipartitions) of the clades in parent nodes. We leave for future work investigating the performance of more complicated SBNs for general trees, structure learning of SBNs, deeper examination of the effect of parameter sharing, and further applications of SBNs to other probabilistic learning problems in tree spaces, such as designing more efficient tree proposals for MCMC transition kernels and providing flexible and tractable distributions for variational inference.

## Acknowledgements

This work supported by National Science Foundation grant CISE-1564137, as well as National Institutes of Health grants R01-GM113246 and U54-GM111274. The research of Frederick Matsen was supported in part by a Faculty Scholar grant from the Howard Hughes Medical Institute and the Simons Foundation.

## Footnotes

[1] The subsplits $S_2, S_3, \ldots$ are well defined once the split in $S_1$ (or equivalently the root) is given.

[2]The same $\alpha$ is used for the real datasets since the sample sizes are roughly the same, although the number of unique trees are quite different.

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
