[Supplementary Material]

# Supplementary Material for Generalizing Tree Probability Estimation via Bayesian Networks

## A  The SBN Representation of The Rooted Tree from Figure 1

Figure A.1: The SBN representation of the rooted tree from Figure 1. For ease of presentation, we use the indexes of the leaf labels for subsplits. We also omit the fake splits after depth 4.

## B  Proof of Lemma 1

*Proof.* It suffices to prove the lemma for the minimum SBN $\mathcal{B}^*_{\mathcal{X}}$. For any rooted tree $T$, with natural indexing, a compatible assignment is easy to obtain by following the splitting process and satisfying the compatibility requirement defined in Definition 2. Moreover, the depth of the splitting process is at most $N - 1$, where $N$ is the size of $\mathcal{X}$. To show this, consider the maximum size of clades, $d_i$, in the $i$-th level of the process. The first level is for the root split, and $d_1 \leq N - 1$ since the subclades need to be proper subset of $\mathcal{X}$. For each $i > 1$, if $d_i > 1$, the splitting process continues and the maximum size of clades in the next level is at most $d_i - 1$, that is $d_{i+1} \leq d_i - 1$. Note that the split process ends when $d_i = 1$, its depth therefore is at most $N - 1$. The whole process is deterministic, so each rooted tree can be uniquely represented as a compatible assignment of $\mathcal{B}^*_{\mathcal{X}}$. On the other hand, given a compatible assignment, we can simply remove all fake splits. The remaining net corresponds to the true splitting process of a rooted tree, which then can be simply restored. $\square$

## C  Proof of Proposition 1

*Proof.* Denote the set of all SBN assignments as $\mathcal{A}$. For any assignment $a = \{S_i = s_i^a\}_{i \geq 1}$, we have

$$p_{\text{sbn}}(a) = p(S_1 = s_1^a) \prod_{i>1} p(S_i = s_i^a | S_{\pi_i} = s_\pi^a) > 0 \Rightarrow a \text{ is compatible.} \tag{1}$$

As a Bayesian network,

$$\sum_{a \in \mathcal{A}} p_{\text{sbn}}(a) = \sum_{s_1} p(S_1 = s_1) \prod_{i>1} \sum_{s_i} p(S_i = s_i | S_{\pi_i} = s_{\pi_i}) = 1.$$

By Lemma 1,

$$\sum_T p_{\mathrm{sbn}}(T) = \sum_{a \sim \mathrm{compatible}} p_{\mathrm{sbn}}(a) = \sum_{a \in \mathcal{A}} p_{\mathrm{sbn}}(a) = 1.$$

<div style="text-align: right;">□</div>

## D   Proof of Proposition 2

*Proof.*

$$\sum_{T^{\mathrm{u}}} p_{\mathrm{sbn}}(T^{\mathrm{u}}) = \sum_{T^{\mathrm{u}}} \sum_{S_1 \sim T^{\mathrm{u}}} p(S_1) \prod_{i>1} p(S_i | S_{\pi_i})$$

$$= \sum_T p_{\mathrm{sbn}}(T)$$

$$= 1.$$

The second equality is due to the one-to-one correspondence between rooted trees and unrooted trees with specific rooting edges (compatible splits):

$$T \overset{\text{1-to-1}}{\Longleftrightarrow} (T^{\mathrm{u}}, S_1), \ S_1 \sim T^{\mathrm{u}}.$$

<div style="text-align: right;">□</div>

## E   Figure for Shared Parent-child Subsplit Pairs in SBNs

Figure E.1: Subsplit pairs shared among different nodes in SBNs. Top panels show the rooted trees that share the same local parent-child subsplit pair $((B, C), D)$. Bottom panels are their corresponding representations in SBNs. Just like in the trees, the same pair could appear at multiple locations in SBNs.

## F   Related Work

The idea of using conditional probabilities associated with local tree structures (e.g., clades) to approximate the whole posterior distribution on tree topologies started with Höhna and Drummond [2012], where the authors extended the additive binary (AB) coding scheme [Farris et al., 1970, Brooks, 1981] to a new algorithm that approximates the posterior probabilities of trees by a product of conditional clade probabilities (CCP). However, the CCP method does not provide a true probability distribution on trees and requires renormalization to obtain the estimated probabilities. Therefore, CCP is not tractable for trees with many taxa except for restricting to a small subset of high probability trees. Larget [2013], further extended the CCP method by allowing joint modeling of all children clades of the parent clade in the conditional clade probabilities. The resulting conditional clade distribution (CCD) approach not only improves the approximation accuracy of CCP but also leads to

a valid probability distribution on the tree space. However, CCD is still not flexible enough to capture the complexity of inferred posterior distributions on real data [Whidden and Matsen, 2015].

SBNs also leverage parameter sharing to reduce the number of free parameters in the model and promote the generalization performance. Similar to weight sharing used in convolutional networks for detecting *translationally-invariant structure* of images (e.g., edges, corners), our heuristic parameter sharing in SBNs is for identifying *conditional splitting patterns* of leaf-labeled trees. Parameter sharing has been widely employed in probability graphical models such as hidden Markov models [Baum et al., 1970], dynamic Bayesian networks [Murphy, 2002], conditional random fields [Lafferty et al., 2001] and statistical relational models [Getoor and Taskar, 2007]. Other than specifying the tied parameters a priori, recent work [Chou et al., 2016, 2018] also utilizes quantization for automatic parameter tying.

# G   On Expectation Maximization for SBNs

Using the conditional probabilities of the missing root node, we can construct the following adaptive lower bound

$$LB^{(n)}(T^{\mathrm{u}};p) = \sum_{S_1 \sim T^{\mathrm{u}}} p(S_1|T^{\mathrm{u}}, \hat{p}^{\mathrm{EM},(n)}) \log \left( \frac{p(S_1) \prod_{i>1} p(S_i|S_{\pi_i})}{p(S_1|T^{\mathrm{u}}, \hat{p}^{\mathrm{EM},(n)})} \right) \le \log L(T^{\mathrm{u}};p). \quad (2)$$

The above adaptive lower bound $LB^{(n)}(T^{\mathrm{u}};p)$ contains a constant term that only depends on the current estimates, and another term that is the expected complete log-likelihood with respect to the conditional distribution of $S_1$ given $T^{\mathrm{u}}$ and the current estimates:

$$Q^{(n)}(T^{\mathrm{u}};p) = \sum_{S_1 \sim T^{\mathrm{u}}} p(S_1|T^{\mathrm{u}}, \hat{p}^{\mathrm{EM},(n)}) \left( \log p(S_1) + \sum_{i>1} \log p(S_i|S_{\pi_i}) \right). \quad (3)$$

Summing (3) over $T^{\mathrm{u}} \in \mathcal{D}^{\mathrm{u}}$ together with conditional probability sharing, we get the complete data score function

$$
\begin{aligned}
Q^{(n)}(\mathcal{D}^{\mathrm{u}};p) &= \sum_{k=1}^{K} Q^{(n)}(T_k^{\mathrm{u}};p) \\
&= \sum_{k=1}^{K} \sum_{e \in E(T_k^{\mathrm{u}})} q_k^{(n)}(S_1 = s_{1,k}^e) \left( \log p(S_1 = s_{1,k}^e) + \sum_{i>1} \log p(S_i = s_{i,k}^e | S_{\pi_i} = s_{\pi_i,k}^e) \right) \\
&= \sum_{s_1 \in \mathbb{C}_r} \overline{m}_{s_1}^{\mathrm{u},(n)} \log p(S_1 = s_1) + \sum_{s|t \in \mathbb{C}_{\mathrm{ch|pa}}} \overline{m}_{s,t}^{\mathrm{u},(n)} \log p(s|t)
\end{aligned}
$$

$$(4)$$

where

$$\overline{m}_{s_1}^{\mathrm{u},(\mathrm{n})} = \sum_{k=1}^{K} \sum_{e \in E(T_k^{\mathrm{u}})} q_k^{(n)}(S_1 = s_1) \mathbb{I}(s_{i,k}^e = s_1)$$

$$\overline{m}_{s,t}^{\mathrm{u},(n)} = \sum_{k=1}^{K} \sum_{e \in E(T_k^{\mathrm{u}})} q_k^{(n)}(S_1 = s_{1,k}^e) \sum_{i>1} \mathbb{I}(s_{i,k}^e = s, s_{\pi_i,k}^e = t)$$

are the expected frequency counts.

## G.1   Proof of Theorem 1

*Proof.*

$$
\begin{aligned}
Q^{(n)}(T^{\mathrm{u}};p) - Q^{(n)}(T^{\mathrm{u}};\hat{p}^{\mathrm{EM},(n)}) &= LB^{(n)}(T^{\mathrm{u}};p) - LB^{(n)}(T^{\mathrm{u}};\hat{p}^{\mathrm{EM},(n)}) \\
&\le \log L(T^{\mathrm{u}};p) - \log L(T^{\mathrm{u}};\hat{p}^{\mathrm{EM},(n)})
\end{aligned}
$$

since $LB^{(n)}(T^{\mathrm{u}}; p) \leq \log L(T^{\mathrm{u}}; p)$ and

$$
\begin{aligned}
LB^{(n)}(T^{\mathrm{u}}; \hat{p}^{\mathrm{EM},(n)}) &= \sum_{S_1 \sim T^{\mathrm{u}}} p(S_1 | T^{\mathrm{u}}, \hat{p}^{\mathrm{EM},(n)}) \log \left( \frac{p(T^{\mathrm{u}}, S_1 | \hat{p}^{\mathrm{EM},(n)})}{p(S_1 | T^{\mathrm{u}}, \hat{p}^{\mathrm{EM},(n)})} \right) \\
&= \sum_{S_1 \sim T^{\mathrm{u}}} p(S_1 | T^{\mathrm{u}}, \hat{p}^{\mathrm{EM},(n)}) \log p(T^{\mathrm{u}} | \hat{p}^{\mathrm{EM},(n)}) \\
&= \log p(T^{\mathrm{u}} | \hat{p}^{\mathrm{EM},(n)}) \\
&= \log \sum_{S_1 \sim T^{\mathrm{u}}} p(T^{\mathrm{u}}, S_1 | \hat{p}^{\mathrm{EM},(n)}) \\
&= \log L(T^{\mathrm{u}}; \hat{p}^{\mathrm{EM},(n)}).
\end{aligned}
$$

$\square$

With Theorem 1, it is clear that

$$
Q^{(n)}(\mathcal{D}^{\mathrm{u}}; p) - Q^{(n)}(\mathcal{D}^{\mathrm{u}}; \hat{p}^{\mathrm{EM},(n)}) \leq \log L(\mathcal{D}^{\mathrm{u}}; p) - \log L(\mathcal{D}^{\mathrm{u}}; \hat{p}^{\mathrm{EM},(n)}).
$$

As before, the maximum estimates of $Q^{(n)}$ have closed form expressions which lead to the following updating formula for the CPDs at the M-step

$$
\hat{p}^{\mathrm{EM},(n+1)}(S_1 = s_1) = \frac{\overline{m}_{s_1}^{\mathrm{u},(n)}}{\sum_{s \in \mathbb{C}_r} \overline{m}_s^{\mathrm{u},(n)}} = \frac{\overline{m}_{s_1}^{\mathrm{u},(n)}}{K}, \quad s_1 \in \mathbb{C}_r \tag{5}
$$

$$
\hat{p}^{\mathrm{EM},(n+1)}(s|t) = \frac{\overline{m}_{s,t}^{\mathrm{u},(n)}}{\sum_s \overline{m}_{s,t}^{\mathrm{u},(n)}}, \quad s|t \in \mathbb{C}_{\mathrm{ch|pa}}. \tag{6}
$$

## G.2 Regularization

For the score function in (4), the conjugate prior distributions have to be Dirichlet distributions

$$
S_1 \sim \mathrm{Dir}(\alpha_{s_1} + 1, s_1 \in \mathbb{C}_r), \quad \cdot|t \sim \mathrm{Dir}(\alpha_{s,t} + 1, s|t \in \mathbb{C}_{\mathrm{ch|pa}})
$$

where $\alpha_{s_1} > 0$, $\alpha_{s,t} > 0$ are some hyper-parameters. The regularized score function, therefore, takes the following form

$$
\begin{aligned}
Q^{\mathrm{R},(n)}(\mathcal{D}^{\mathrm{u}}; p) &= Q^{(n)}(\mathcal{D}^{\mathrm{u}}; p) + \sum_{s_1 \in \mathbb{C}_r} \alpha_{s_1} \log p(S_1 = s_1) + \sum_{s|t \in \mathbb{C}_{\mathrm{ch|pa}}} \alpha_{s,t} \log p(s|t) \\
&= \sum_{s_1 \in \mathbb{C}_r} (\overline{m}_{s_1}^{\mathrm{u},(n)} + \alpha_{s_1}) \log p(S_1 = s_1) + \sum_{s|t \in \mathbb{C}_{\mathrm{ch|pa}}} (\overline{m}_{s,t}^{\mathrm{u},(n)} + \alpha_{s,t}) \log p(s|t).
\end{aligned} \tag{7}
$$

One can adapt the hyper-parameters according to this decomposition as follows

$$
\alpha_{s_1} = \alpha \cdot \tilde{m}_{s_1}^{\mathrm{u}}, \quad \alpha_{s,t} = \alpha \cdot \tilde{m}_{s,t}^{\mathrm{u}}
$$

where $\alpha$ is the global regularization coefficient that balances the effect of the data on the estimates in comparison to the prior, and $\tilde{m}_{s_1}^{\mathrm{u}}, \tilde{m}_{s,t}^{\mathrm{u}}$ are the equivalent sample counts for the root splits and parent-child subsplit pairs that distribute the regularization across the CPDs. In practice, one can simply use the the sample frequency counts as equivalent sample counts as we did in our experiments.

**Remark:** The choice of hyper-parameter setting is not throughly studied and what we do here is some modification from the common practice in Bayesian estimation for learning Bayesian networks; see Buntine [1991] for more details.

Theorem 1 also implies that maximizing (or improving) the regularized $Q$ score is sufficient to improve the regularized log-likelihood. Denote the regularization term as

$$
R(p) = \sum_{s_1 \in \mathbb{C}_r} \alpha \tilde{m}_{s_1}^{\mathrm{u}} \log p(S_1 = s_1) + \sum_{s|t \in \mathbb{C}_{\mathrm{ch|pa}}} \alpha \tilde{m}_{s,t}^{\mathrm{u}} \log p(s|t)
$$

we have

$$Q^{\mathrm{R},(n)}(\mathcal{D}^{\mathrm{u}};p) - Q^{\mathrm{R},(n)}(\mathcal{D}^{\mathrm{u}};\hat{p}^{\mathrm{EM},(n)}) = Q^{(n)}(\mathcal{D}^{\mathrm{u}};p) - Q^{(n)}(\mathcal{D}^{\mathrm{u}};\hat{p}^{\mathrm{EM},(n)}) + R(p) - R(\hat{p}^{\mathrm{EM},(n)})$$
$$\leq \log L(\mathcal{D}^{\mathrm{u}};p) + R(p) - \big(\log L(\mathcal{D}^{\mathrm{u}};\hat{p}^{\mathrm{EM},(n)}) + R(\hat{p}^{\mathrm{EM},(n)})\big).$$

Similarly, the regularized score $Q^{\mathrm{R},(n)}$ can be maximized in the same manner as $Q^{(n)}$

$$\hat{p}^{\mathrm{EM},(n+1)}(S_1 = s_1) = \frac{\overline{m}_{s_1}^{\mathrm{u},(n)} + \alpha \tilde{m}_{s_1}^{\mathrm{u}}}{\sum_{s_1 \in \mathbb{C}_r}(\overline{m}_{s_1}^{\mathrm{u},(n)} + \alpha \tilde{m}_{s_1}^{\mathrm{u}})} = \frac{\overline{m}_{s_1}^{\mathrm{u},(n)} + \alpha \tilde{m}_{s_1}^{\mathrm{u}}}{K + \alpha \sum_{s_1 \in \mathbb{C}_r} \tilde{m}_{s_1}^{\mathrm{u}}}, \quad s_1 \in \mathbb{C}_r \quad (8)$$

$$\hat{p}^{\mathrm{EM},(n+1)}(s|t) = \frac{\overline{m}_{s,t}^{\mathrm{u},(n)} + \alpha \tilde{m}_{s,t}^{\mathrm{u}}}{\sum_s(\overline{m}_{s,t}^{\mathrm{u},(n)} + \alpha \tilde{m}_{s,t}^{\mathrm{u}})}, \quad s|t \in \mathbb{C}_{\mathrm{ch|pa}}. \quad (9)$$

### G.3 Computational Complexity

Compared to those for rooted trees, the expected frequency counts $m_{s_1}^{\mathrm{u}}, m_{s,t}^{\mathrm{u}}, \overline{m}_{s_1}^{\mathrm{u},(n)}, \overline{m}_{s,t}^{\mathrm{u},(n)}$ for unrooted trees involves additional summation over all the edges. However, we can precompute all the accumulated sums of the root probabilities on the edges in a postorder tree traversal for each tree and the computation cost remains $\mathcal{O}(KN)$. Each E-step in EM would in general cost $\mathcal{O}(KN^2)$ time since for each tree, $\mathcal{O}(N)$ probability estimations are required and each takes $\mathcal{O}(N)$ time. Notice that most of the intermediate partial products involved in those SBN probability estimates are shared due to the structure of trees, we therefore use a two-pass algorithm, similar to the one used in [Schadt et al., 1998], that computes all SBN probability estimates for each tree within two loops over its edges. This reduces the computational complexity of each E-step to $\mathcal{O}(KN)$. Overall, the computational complexities of maximum lower bound estimate and each EM iteration are both $\mathcal{O}(KN)$, the same as CCD and SBNs for rooted trees.

## H Run Time Comparison

In this section, we present a runtime comparison of different algorithms on simulated data with varying $K$ and $\beta$. We see that SBN based methods significantly improve the approximation accuracy of CCD. With additional time budget, sbn-em and sbn-em-$\alpha$ further improve the performance of sbn-sa, and regularization is useful especially for diffuse distributions with limited sample size (Figure H.1). All experiments were done on a 2016 MacBook Pro (2.9 GHz Intel Core i5).