[Reviews · NeurIPS 2018]

Reviewer 1



In this paper the authors propose an efficient method for tree probability estimation (given a collection of trees) that relies on the description of trees as subsplit Bayesian networks. Through this representation, the authors relax the classic conditional clade distribution - which assumes that given their parent, sister clades are independent - and assume instead that given their parent subsplit, sister subsplits are independent, thus allowing more dependence structure on sister clades. The authors first present a simple maximum likelihood estimation algorithm for rooted trees, and then propose two alternatives to generalize their work to unrooted trees. They finally illustrate their method on both simulated and real-data experiments. I think this paper is very well written, in particular I have greatly appreciated the Background and SBN description sections that make use of a simple though not trivial example to introduce new notions and provide useful insights on the assumptions. The authors might want to add a supplemental figure describing the representation of the rooted tree from figure 1 in terms of SBN to further illustrate this notion. To my knowledge the use of SBN for posterior distribution of trees inference is knew, and the results are certainly encouraging. However, if I understand correctly; the ultimate goal is to reconstruct the posterior distribution of the shape of the tree. The authors are particularly interested in phylogenetic trees, for which the size of the edges is a crucial information as it represents the age of the most recent common ancestor. Do the authors have some insight on how to extend SBN to these kind of trees? As additional minor points, I would appreciate if the authors could provide details on the KL computation with Dirichlet prior (as a supplementary material). I also think the authors should provide the runtime required by each algorithm in the experimental study. Finally, in the introduction, I am not sure of what the authors mean by "SFR foes not support trees beyond observed samples", which seems to me is the case for all presented methods including SBM. Have I misunderstood something? -- I thank the authors for their response and think they have answered my questions in an appropriate manner. I look forward to reading their updated submission and extension, and think this paper should be accepted.

Reviewer 2



In this paper, the authors propose a framework for leaf-labeled tree probability estimation given a collection of trees such as MCMC samples by introducing a novel structure called subsplit bayesian networks (SBN) which relaxes the assumption that clades are conditionally independent. They show that every tree can be uniquely represented as a set of subsplits. They compute maximum likelihood with weight sharing to learn the parameters of SBNs for rooted leaf-labeled trees. They also discuss that how the proposed framework can be generalized for unrooted trees and propose variational EM algorithms for this case. The paper is generally clearly written. The model is well-defined, with examples making the relaxation of independence clear. Also in the learning progress, the update steps are carefully derived. The paper could benefit doing the following additional experiments:1. One major contribution of the proposed model is its flexibility. Compared to CCD, it should be able to gain more accuracy while there are some more dependence on the parameters. Drawing the parameters with different degree of diffusion doesn't really vary the depency. Thus the experiments the author showed are not sufficient to conclude how much is the gain from the relaxation of independence assumption. Also, it is surprising that as \beta increases and the parameters are more diffused, SBN is gaining less accuracy compared to CCD(figure 4, third picture from left). By using the shared parameter technique, when the parameters are more diffused this sharing parameter approximation intuitively is more accurate. 2. It would be useful if there are some experiments benchmarking the computational and statistical cost of parameter sharing especially given that parameter sharing entails a trad eoff in variance for some bias. 3. In section 3, B^*_x is claimed to be the minimal SBN over X. The proof seems to be straightforward. However, the proof is needed to support this statement. 4.In section 3, it is shown that for a particular example, CCD can be viewed as a simplification of SBNs by further approximation of the conditional probabilities. Does this hold in general? Minor comments: 1. Some of the parameters should be carefully defined. E.g., in section 4.2 the unrooted tree T^u is not defined. 2. In section 3, it is not clear that how lexicographical order is defined on clades for the tree in Figure 1. According to the definition of the subsplit, if (Y,Z) is a subsplit then Y>Z. In the example ({O_3},C_6) and (C_7,{O_8}) are two subsplits; therefore, we have {O_3} > {O_4,O_5} and {O_6,O_7}>{O_8} in lexicographical order which is not clear how t his order is defined.

Reviewer 3



The paper proposes to model the distribution over phylogenetic trees using a special kind of Bayesian networks called subsplit Bayesian networks(SBNs). SBN nodes can be considered as categorical random variables taking on values over possible subsplits which are partitions of the labels in the domain. Conditional probabilities in SBNs model the probability of a child node taking on a subsplit value conditioned on its parent(s) subsplit value(s). Given an SBN structure and a sample of rooted or unrooted trees, the authors use standard procedures (closed form MLE estimations for rooted trees and EM for unrooted trees) to estimate the parameters (conditional probabilities) of the SBN. The authors adopt a similar method of weight sharing in convolutional neural networks to share parameters in SBNs. The probabilities of phylogenetic trees approximated using SBNs have been shown to be more accurate compared to other methods on both synthetic and real data. I am not an expert on phylogenetics. I am familiar with Bayesian network representation, inference and learning (both with observed and latent variables). Hence, I could understand the idea of modeling the probability distribution over possible trees using SBNs. There is still room for improvements (see comments below), but compared to current methods like SRF and CCD, SBNs are more flexible and general. Hence, I am voting for an accept. Quality: The more challenging and interesting task of structure learning of SBNs is yet to be explored. Simple tree like SBNs have been considered in the paper although it has been mentioned in section 3 that more complicated dependencies may be introduced. More recent works than [LeCun et al. 1998] which aims to reduce the number of parameters in a model by parameter tying are by Chou, Li, et al. "On Parameter Tying by Quantization." (AAAI. 2016), and "Automatic Parameter Tying: A New Approach for Regularized Parameter Learning in Markov Networks." (AAAI 2018). I would suggest the authors to explore such works on parameter sharing in pgms. Also, no experimental evaluation was done to see the reduction in the number of parameters through sharing in the learned models. The paper lacks a related works section. The authors might consider adding one to SM. Clarity: Overall, the paper was not very difficult to understand. The following questions are related to the experimental section. It is not clear why \alpha was fixed in experiments with real datasets. Is any smoothing performed when estimating the parameters in the CCD/SBN CPTs? Also, what was the average number of iterations performed during EM? The reason behind the very low number of sampled trees for DS2 and DS3 should be explained. In [Larget, 2013] a much greater number of samples were generated for both the datasets. General comments regarding notations: Subscript k missing for parent assignment in L(T_k) in line 130 Superscript (n) in EM subsection not clearly introduced. Originality and significance: Learning the distribution over trees using a generative probabilistic model like SBN has advantages over simple SRF and CCDs. I believe other interesting probabilistic queries can be carried out on these distributions besides computing tree probabilities. ---------------------------------------------------------------------------------------------------------- I thank the authors for answering to the reviewers questions. The authors have tried to answer all the questions, based on which I am updating the score for the paper. I would suggest the authors to include experimental results on parameter sharing in the current paper.